# Effect of Plyometric versus Ankle Stability Exercises on Lower Limb Biomechanics in Taekwondo Demonstration Athletes with Functional Ankle Instability

**DOI:** 10.3390/ijerph17103665

**Published:** 2020-05-22

**Authors:** Ha Min Lee, Seunghue Oh, Jung Won Kwon

**Affiliations:** 1Department of Physical Therapy, College of Health Sciences, Dankook University, 119, Dandae-ro, Dongnam-gu, Cheonan-si, Chungcheongnam-do 31116, Korea; ptmin3215@naver.com; 2Department of Physical Therapy, Graduate School, Dankook University, 119, Dandae-ro, Dongnam-gu, Cheonan-si, Chungcheongnam-do 31116, Korea; rock3224@naver.com

**Keywords:** motion analysis, taekwondo, plyometric exercise, stability exercise, functional ankle instability, sports injury

## Abstract

Background: This study aimed to compare the effects of plyometric and ankle stability exercises on the dynamic balance and lower limb kinematic and kinetic parameters of Taekwondo demonstration athletes with functional ankle instability. Methods: Fourteen subjects participated in this study and were randomly divided into two groups: a plyometric exercise group (*n* = 7) and an ankle stability exercise group (*n* = 7). Exercises were performed twice a week for 8 weeks. A Y-balance test was used to measure dynamic balance, and a motion analysis system and force plate were used to collect kinematic and kinetic parameters during single-leg drop landing. A paired t-test was used for intragroup comparisons, and an independent t-test was used for intergroup comparisons. Results: In both groups, exercise increased dynamic balance and shock absorption and reduced postural sway on the anteroposterior displacement (*p* < 0.05). The plyometric exercise group decreased their ankle dorsiflexion and increased their knee and hip joint flexion at maximum knee flexion (*p* < 0.05). In contrast, the stability exercise increased their ankle plantar flexion at initial contact (*p* < 0.05). Conclusions: The plyometric exercise group altered their landing strategies using their knee and hip joints to control ankle instability at landing. This study suggests that the application of plyometric exercises in ankle rehabilitation would improve stability and shock absorption and help prevent injuries during Taekwondo demonstrations.

## 1. Introduction

Taekwondo demonstrations are impressive performances of athleticism and artistry and consist of basic Taekwondo skills, such as traditional forms and patterns (called “Poomsae”), choreographed self-defense sequences, and board breaking. Taekwondo demonstrations always require high-intensity tasks, such as jump landing, kicking, breaking, and turning kicks [1,2], which require agility, speed, flexibility, strength, and endurance [1,3,4]. Injuries in Taekwondo demonstration athletes commonly involve the lower limbs [1] and usually occur during jump landing [5].

Depending on the jump-landing height, at least 10 times body weight is applied to the ankles, which creates excessive shock at the joints [6]. Ideally, a landing should involve both legs, but single-leg landings are more common during Taekwondo demonstrations, and most injuries occur during single-leg landing tasks [5]. Thus, the load applied to an ankle may cause ankle injury, and repeated injuries may progress to functional ankle instability (FAI) [7,8], which is a type of chronic ankle instability that leads to loss of proprioception and neuromuscular changes [7,8], muscle weakness [7], and lower limb misalignment [9]. These cause dynamic changes in the affected ankle joints, increase risk of injury, and reduce athletic performance [10].

Ankle rehabilitation exercises are designed to improve flexibility, range of motion, muscle strength, proprioception, and neuromuscular control and prevent FAI [11,12]. Traditional ankle rehabilitation exercises consist of range of motion exercises [11,13], progressive strength training [12,14], proprioceptive exercises [11,15], and activity-specific training [12]. A previous study reported that a 6-week plyometric exercise course improved the functional performance of an athlete with an ankle sprain [16], and other studies have shown that plyometric exercise improves proprioception, strength, and response speed in patients with acute lateral ankle sprain [17] and reduces postural sway during exercise in patients with ankle instability [16].

Repeated landings could negatively affect sport performance and increase injury risk [18]. Plyometrics is a type of high-intensity exercise that reinforces muscle strength through repeated concentric and eccentric contractions [17]. Although the effects of plyometric exercise have been investigated on several occasions, its effects on the kinematic and kinetic parameters of Taekwondo demonstration athletes with FAI have not been established.

Based on the hypothesis that plyometric exercise can promote ankle stability by changing landing strategies, the purpose of this study is to investigate the effects of plyometric and ankle stability exercises on dynamic balance and the lower limb kinematic and kinetic parameters of Taekwondo demonstration athletes with functional ankle instability.

## 2. Materials and Methods

### 2.1. Subjects

Taekwondo demonstration athletes were voluntarily recruited in the local university, and only 14 volunteers suitable for this study participated. The inclusion criteria were as follows: no history of a congenital orthopedic deformity in the lower limbs, evaluated as FAI (ankle instability instrument score >5), no orthopedic surgery during the previous 6 months, and no vestibular or balance disorder. Note that we enrolled the athletes with mild functional ankle instability to promote the functional performance that maintains static and dynamic balance and kinesthetic control. All subjects understood the purpose of the study and voluntarily signed the subject consent form to participate in the study. The study protocol was approved by the institutional review board of the local university (Institutional Review Board of Dankook University—DKU 2019-04-022). Demographic and clinical data were collected, and there was no significant difference between the groups (*p* > 0.05) (Table 1).

### 2.2. Measurements

#### 2.2.1. Y-balance Test (YBT)

The YBT was developed to predict injury and measure dynamic balance, flexibility, and proprioception [10]. The YBT protocol was designed to reach the anterior (ANT), posterior–lateral (PL), and posterior–medial (PM) sides on a single leg [10,19]. The PL and PM sides are located 135° from the ANT side [19]. In a standing position, the subject held his or her pelvis, maintained balance on a single leg, and then reached with the other leg as much as possible in three directions. The side affected by FAI was set as the weight-bearing axis. The inter- and intrarater reliability of the YBT was found to be excellent [14]. To compensate for leg length differences, we used normalized composite scores (CSs), which were determined by summing the reach distances in each direction, dividing the result by three times the leg length and then multiplying by 100 to obtain percentages [14]. Subjects were aware that the following were considered failures: a weight-bearing foot fall from the center point, foot dragging to increase reach distance, failure to return the starting position, and the use of hands to maintain balance [10,14]. Measurements were repeated until three sets of successful data were obtained.

#### 2.2.2. Kinematic and Kinetic Analysis

A 3D motion analysis system with six cameras (Qualisys System, Qualisys AB, Gothenburg, Sweden) was used to record the kinematic and kinetic parameters of the lower limb during single-leg drop landing. Two force plates (Advanced Mechanical Technology, Inc., Watertown, MA, USA) were used to measure the ground reaction forces (GRFs) and determine the pressure centers during landing. Twenty-eight markers (super-spherical markers, Qualisys AB, Gothenburg, Sweden) were attached to specific anatomical landmarks of the lower limbs, as described by Helen Hayes [20]. Cameras were set at 100 frames/s with a shutter speed of 1/500 s. The cutoff frequency used to reduce noise was set at 6 Hz. Two force plates were connected to a sync LED for image analysis and synchronization, and a Qualisys A/D board was used for time synchronization. The landing force were collected at 400 Hz.

All subjects performed single-leg drop landings from a 45-cm-high box; the distance between the box and force plate was set at 20 cm [21]. Subjects wore short stretch pants and were instructed to fold their arms over their chests to limit upper limb movement. Subjects were instructed to step (not jump) off the box and naturally land on one-foot; they were aware that the following were considered failures: performing the landing by pushing on the box or jumping to the force plate, an inability to maintain balance, an additional touch on the force plate after landing, and the arm falling off the chest during landing [9]. Measurements were repeated until three successful datasets were obtained.

### 2.3. Interventions

The 14 volunteers were randomly allocated to the plyometric exercise (PE) group or ankle stability exercise (ASE) group. The 8-week training program was performed twice a week for 1 h. Subjects conducted warm-up and cool-down exercises for 10 min; thus, the training program lasted for 40 min.

The PE program consisted of hopping on different sides, jumping with or without a box, and performing single-leg and double-leg jumping (Table 2) [17,22]. The exercise involved three sets of eight repetitions. The ASE program consists of using balance pads and performing elastic band and active range of motion exercises [12]. The exercise involved three sets of 15 repetitions. Each exercise repetition was set according to the Borg scale. The rest time between sets was 30 s. Subjects were required to perform exercises slowly and accurately. After 4 weeks of exercise, exercise intensity, volume, sets, and session frequency were adjusted based on the subject’s Borg Scale score. All subjects were under the instruction and guidance of a skilled supervisor during the exercises. Data were collected before (pretest) and after 8 weeks of exercise (posttest).

### 2.4. Data Processing

A visual 3D system (Visual3D v6 Professional, C-Motion, Inc., Germantown, MD, USA) was used to edit and digitize the motion capture images and process the kinematic and kinetic parameters of the single-leg drop landing.

The following events were used to analyze the single-leg drop landings: (1) initial contact (IC), the first time the foot touched the force plate and GRF >5 N; and (2) the maximum knee flexion (MKF), the moment of the maximum flexion of the knee joint after IC.

Kinematic parameters were analyzed for hip, knee, and ankle joint angles in the sagittal plane. The kinetic parameters collected were the center of pressure (COP) displacement and vertical GRF (VGRF). COPs were classified as COP X (anterior-posterior direction) and COP Y (medial-lateral direction), which were used to determine posture stability at landing. VGRFs were measured in the vertical direction (*z*-axis) and normalized by dividing the values by body weight.

### 2.5. Statistical Analysis

Data analysis was performed using the SPSS software (version 21.0, SPSS Inc., Chicago, IL, USA). A Shapiro–Wilk test was used to test normality. A paired t-test was used to determine the significance of the differences between the pre- and post-test values within the PE and ASE groups. Additionally, an independent t-test was used for the intergroup analysis between the PE and ASE groups. *p*-values < 0.05 indicate statistical significance.

## 3. Results

Changes in the dynamic balance and kinematic and kinetic parameters in the two groups are shown in Table 3. Before intervention, all pre-test values (base line) showed no significant differences between the two groups (*p* > 0.05). The average CS was calculated to assess dynamic balance. The CS was significantly increased in both groups (*p* < 0.05). The ASE group had a significantly higher CS compared to the PE group (*p* < 0.05) (Table 3).

In the PE group, the ankle joint angle at IC was not significantly different between the pre- and posttests (*p* > 0.05), but in the ASE group, the ankle joint angle at IC was significantly higher in the posttest (*p* < 0.05). In both groups, the knee and hip joint angles at IC were not significantly different between the pre- and posttests (*p* > 0.05). The ankle, knee, and hip joint angles at MKF were significantly different in the PE group (*p* < 0.05), but there were no significant differences in the ASE group (*p* > 0.05). In the between-group analysis, the ankle joint at IC experienced significantly greater plantar flexion (*p* < 0.05), while the ankle joint at MKF was significantly more dorsiflexed in the ASE group than in the PE group (*p* < 0.05). Differences in the knee joint angles at both IC and MKF showed that the PE group had significantly greater flexion compared to the ASE group (*p* < 0.05). However, there were no significant differences in hip joint angle at both the IC and MKF between the groups (*p* > 0.05).

In the PE group, COP X, IC VGRF, and MKF VGRF were significantly lower in the posttest (*p* < 0.05). In the ASE group, COP X and IC VGRF were significantly lower in the posttest (*p* < 0.05), but MKF VGRF was not significantly different between the pre- and post-tests (*p* > 0.05). In both groups, COP Y was not significantly different in the posttest (*p* > 0.05). In the between-group analysis, there were no significant differences in COP X, COP Y, and MKF VGRF (*p* > 0.05). However, IC VGRF was significantly lower in the ASE group (*p* < 0.05).

## 4. Discussion

The aim of this study was to investigate the effects of PE and ASE on the dynamic balance and lower limb kinematic and kinetic parameters of Taekwondo demonstration athletes with FAI. PE and ASE have positive effects on Taekwondo demonstration athletes with FAI, but the two groups showed different compensatory patterns after 8 weeks training. The main findings of this study are as follows: After training, the PE group showed a decrease in ankle joint angle but increases in knee and hip joint angles in MKF, whereas the ASE group showed an increase in ankle joint angle at IC. Both groups showed improved dynamic balance and shock absorption.

Previous studies reported that FAI is associated with a lack of dynamic balance and that this lack of dynamic balance is the cause of ankle re-damage [23,24]. Injury risk in athletes is increased by reductions in strength, flexibility, neuromuscular control, and proprioception [10]. In the present study, there was a significant increase in the CS from pretest to posttest in both groups. Increases in CS indicate improvement in dynamic balance. Changes in dynamic balance are caused by changes in proprioception and neuromuscular control [8]. Previous studies reported that PE could contribute to improvements in vertical jumping, acceleration, leg strength, muscular power, joint awareness, and overall proprioception [25,26]. Additionally, Mattacola and Dwyer [27] reported that an ASE consisting of ankle muscle strengthening and proprioceptive sense exercises was effective in developing dynamic balance. However, dynamic balance was significantly improved in the ASE group compared to the PE group, according to our results. A study conducted by Hoch et al. [28] reported that the weight-bearing ankle dorsiflexion range of motion is significantly correlated with the performance of anterior reach distance. Additionally, subjects with chronic ankle instability exhibited less knee flexion associated with jump landing, which may be related to limited weight-bearing ankle dorsiflexion [29]. Therefore, in the present study, the ASE improved weight-bearing ankle dorsiflexion, leading to changes in dynamic balance.

Ankle range of motion is an important factor in landing mechanics and functional movement patterns. Subjects with FAI have been shown in previous studies to have abnormal landing patterns due to dorsi- and plantar flexion limitations and reduced ankle joint range of motion [18,30]. During IC on landing, the plantar flexion angle is particularly important because it reduces GRF by increasing the ankle moment arm [31]. In the present study, the ASE group achieved greater plantar flexion angles than the PE group in the posttest. It is thought that the ankle joint angle recovery is achieved by proprioception and muscle strength exercises, which increase flexibility, position sense, and neuromuscular control. Unlike PE, which focuses on training the knee and hip joints, ASE focuses on ankle joints and, thus, has a greater effect on the ankle joint angle at IC. ASE seems to promote an effective landing strategy by restoring ankle muscle strength and joint range of motion. In contrast, PE promotes a more efficient landing strategy in MKF.

In the present study, the PE group seemed to use knee and hip joints more commonly to control ankle instability during landing than the ASE group. Huang et al. [16] reported that knee and hip flexion angles increased after plyometric training in a drop-landing task. In other studies, increased knee flexion and hip flexion during the jump–landing task enabled the body to absorb joint forces more effectively and promote the mechanical advantage of soft tissue structures that provide joint stability [32,33]. Flexion of the lower limb joints during landing is required to control the amount of the moment caused by body mass and gravitational acceleration [34]. Indeed, knee joint movement is the strongest shock-absorbing mechanism, and the knee joint’s angle limitation increases the risk of injury [35]. PE desensitizes the Golgi tendon organs through a stretch-shortening exercise, which allows the elastic components of the muscles to better tolerate stretching [16]. Moreover, PE includes repeated jump landing, which optimizes hamstring activation; this leads to greater knee and hip joint flexion than achieved by ASE [32]. In addition, our results showed that knee and hip flexion increased after PE, but ankle dorsiflexion decreased in a single-leg drop landing. These results suggest that reduced ankle dorsiflexion may be due to the COP line moving anteriorly. When the COP is translated anteriorly, the talocrural joint is passively dorsiflexed and then actively plantar flexed to restore balance. Active plantar flexion moves the COP posteriorly so that the line of gravity is moved anteriorly to maintain balance [36]. These different kinematic patterns of the two groups were found to affect the kinetic factors at landing.

In the present study, the anteroposterior sway (COP X) significantly decreased in the two groups, but no significant difference was observed between the two groups. It is thought that both PE and ASE induce muscular strengthening and the neural adaptations that promote functional performance to maintain static and dynamic balance and kinesthetic control. Paterno et al. [37] reported that a comprehensive neuromuscular protocol combined with dynamic balance and plyometric training improved postural sway in the anteroposterior direction. Salehzadeh et al. [38] revealed that using a combinational program (plyometric, technical, balance, and strength exercises) can improve anteroposterior balance. Furthermore, the ASE is a weight-bearing program, which is considered a form of a closed kinetic chain (CKC) exercise in the present study. CKC exercise stimulates the ankle joints and muscle mechanoreceptors and facilitates the co-activation of agonists and antagonists [6]. Therefore, strengthening the ankle’s musculature to give it greater proprioceptive and kinesthetic feedback leads to improvement in postural control and balance. However, our results showed no significant differences in mediolateral sway (COP Y) in the PE and ASE groups. Myer et al. [33] found that plyometric training and dynamic stabilization training decreased the mediolateral sway during single-leg hop landings on the dominant side of healthy women. This may be due to the nature of the exercises in their study. The movement in the mediolateral direction is mainly controlled by the subtalar joint, whereas the movement in the anteroposterior direction is more regulated by the talocrural joint. In this study, the results may be due to the nature of exercises: The reduction in postural sway occurred at the talocrural joint rather than at the subtalar joint [36].

Our results showed that the IC GRF decreased in both groups, but that of the ASE group was lower than that in the PE group at posttest. Generally, the initial shock absorption of the lower extremity is controlled by the ankle joint [6]. ASE improves ankle control during landing by restoring the position sense of the foot. Gutierrez et al. [39] reported that the ankle controls movement by activating the musculature surrounding the joint before landing to control dynamic stability. The dorsiflexion landing shocks are then transmitted to non-contracted tissues (bones, cartilage, ligaments, etc.), while plantar flexion landing shocks are absorbed by the muscles, resulting in a lower GRF. [40] Additionally, during a single-leg drop landing, the peak GRF or peak loading rate is minimized in weight acceptance when the muscle length–tension curve is in the optimal range (plantar flexion angles between 20° and 30°) [31]. In our results, the ASE group showed that the mean plantar flexion angle after exercise was increased, which supports the findings of previous studies on the ideal ankle angle at IC. In MKF, the VGRF was significantly decreased in the PE group but not in the ASE group. These results are related to the increased knee and hip muscle strength gains through PE. After the first ground touch, the shock is transferred to the knee and hip joints [6]. Zhang et al. [6] reported that the proximal muscles of the lower extremities tend to be larger in volume than the distal muscles because the former have relatively shorter tendons, longer muscle fibers, and greater cross-sectional areas. This muscle arrangement, which provides the energy generated before landing contact, can be transported from the distal end to the proximal and larger muscle groups for further dissipation during impact. In addition, a soft landing, which is defined by >90° knee flexion after landing [41], promotes the mechanical advantages of soft tissue structures and provides joint stability [30]. In the present study, the PE program increased the knee flexion angle, allowing for a soft landing. We observed that patients with FAI selected a landing strategy that minimizes the load on the ankle and provides joint stability to compensate for ankle instability.

The present study has several limitations that warrant consideration. First, it is difficult to generalize the effects of training because subjects were limited to a small number of young male Taekwondo demonstration athletes. Thus, our results cannot be applied to female athletes due to biomechanical differences, and a future study must be conducted on a larger sample size. Second, this study was conducted during the regular season, and exercises other than the intervention were not controlled. We recommend that future studies be conducted during the preseason to limit other exercises. Finally, the present study did not address the persistence of the effects of ASE and PE.

## 5. Conclusions

We found that PE and ASE reinforced dynamic balance and decreased postural sway [32,39]. Moreover, the PE group adopted a different landing strategy using knee and hip joints to control ankle instability at landing. Whether these different landing strategies influence complete recovery was not determined in the present study. Nevertheless, our findings suggest that, if plyometric exercise is applied to ankle rehabilitation, the stability and shock absorption during landing can be enhanced and reduce the risk of the sports-related injuries that occur during landing.

## Figures and Tables

**Table 1 ijerph-17-03665-t001:** General characteristics of the subjects.

	PE	ASE	*p*
Age (yrs)	22.00 (1.73)	23.57 (1.62)	0.105
Weight (kg)	69.57 (4.00)	66.57 (8.94)	0.433
Height (cm)	172.43 (5.25)	173.14 (7.78)	0.844
Leg length (cm)	90.56 (3.03)	91.02 (4.42)	0.820
AII (score)	7.14 (1.35)	8.00 (1.53)	0.287

Mean (SD), PE: plyometric exercise, ASE: ankle stability exercise, AII: ankle instability instrument.

**Table 2 ijerph-17-03665-t002:** The 8 week exercise program for each group.

Exercise	1–4 Weeks		5–8 Weeks	
	Program A	Borg Scale	Program B	Borg Scale
Plyometric exercise	X hops (right/left)	5	Squat jumps	7
	Forward hops		Split squat jumps	
	Forward zigzag hops		X hops (right/left)	
	Skater hops		Ankle hops	
	Lateral shuffle		Side hop over box	
	Step up jump on box		Depth drop jump	
	Step up to reverse lunge (right/left)		Standing long jump with box jump	
Ankle stability exercise	Star excursion exercise		Star excursion exercise ^a^	
	Elastic band exercise		Elastic Band exercise ^b^	
	Lower limbs squat		Lower limbs squat	
	Calf raise		Calf raise ^b^	
	Short foot exercise		Short foot exercise	
	Balance Pad leg standing		Balance Pad leg standing ^a^	
	Balance Pad		Balance Pad	
	One leg standing		one leg standing ^a^	
	Balance Pad Lunge		Balance Pad lunge to kick ^a^	
	Balance Pad standing		Balance Pad standing ^c^	

^a^ combined with eye closed, ^b^ combined with resistance, ^c^ combined with catch ring.

**Table 3 ijerph-17-03665-t003:** The data for the CS, kinematic and kinetic parameters during the single-leg drop landing before exercise and after exercise.

	Group	Pre-Test	Post-Test	Difference
CS (%)	PE	99.97 (5.54)	108.91 (3.69) *	8.94 (3.17)
	ASE	92.79 (10.20)	108.03 (6.24) *	15.23 (6.26) †
IC Ankle plantar flexion ROM (°)	PE	22.33 (4.85)	19.69 (4.92)	−2.12 (3.49)
	ASE	24.80 (6.40)	31.56 (7.39) *	6.54 (5.70) †
IC Knee ROM (°)	PE	26.90 (3.52)	32.21 (4.13)	5.35 (5.95) †
	ASE	23.21 (5.77)	22.72 (7.49)	−0.64 (3.84)
IC Hip ROM (°)	PE	24.90 (4.95)	23.66 (6.49)	−1.24 (4.80)
	ASE	24.53 (5.56)	24.99 (5.73)	0.46 (1.94)
MKF Ankle dorsiflexion ROM (°)	PE	20.08 (4.97)	18.04 (5.80) *	−2.04 (2.03)
	ASE	15.36 (3.34)	20.45 (5.55)	5.09 (6.93) †
MKF Knee ROM (°)	PE	84.07 (5.10)	92.80 (7.32) *	8.74 (7.02) †
	ASE	77.21 (12.93)	78.05 (13.78)	0.84 (5.75)
MKF Hip ROM (°)	PE	52.78 (8.79)	55.45 (7.95) *	2.67 (2.30)
	ASE	51.31 (9.22)	52.42 (9.50)	1.10 (2.57)
COP X (mm)	PE	10.63 (1.07)	9.29 (1.01) *	−1.34 (1.30)
	ASE	10.11 (1.23)	8.99 (0.50) *	−1.11 (0.90)
COP Y (mm)	PE	16.86 (4.88)	14.47 (1.92)	−2.39 (4.03)
	ASE	14.40 (5.40)	14.07 (2.14)	−0.32 (0.92)
IC VGRF (N/kg)	PE	664.44 (106.13)	581.32 (120.11) *	−83.12 (41.58)
	ASE	703.69 (91.53)	536.57 (109.88) *	−167.13 (75.13) †
MKF VGRF (N/kg)	PE	776.92 (64.34)	732.81 (84.12) *	−44.11 (43.70)
	ASE	755.66 (116.15)	697.02 (106.08)	−58.63 (73.27)

Means (SD), ASE: ankle stability exercise, PE: plyometric exercise, CS: composite score, IC: initial contact, MKF: maximum knee flexion, ROM: range of motion, COP: center of pressure, VGRF: vertical ground reaction force, * significant difference within group (*p* < 0.05), † significant difference between group (*p* < 0.05).

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
