# Peer review of "Effect of Plyometric versus Ankle Stability Exercises on Lower Limb Biomechanics in Taekwondo Demonstration Athletes with Functional Ankle Instability"

_ijerph, 2020, doi:10.3390/ijerph17103665_

Round 1

Reviewer 1 Report

1. The aim is clear; it is clear what the study found. But in line 15; (athletes with functional ankle instability) contradictory with line 66.
2. The authors mentioned to some abbreviations in the abstract section. Please, use the abbreviations only in the manuscript sections.
3. Line 24; please don't use any reference in the abstract section.
4. The findings of this study are crucial to ankle rehabilitation, stability and shock absorption during landing, the title is informative and relevant, and the references are relevant, referenced correctly and appropriate key studies are included, however, some references are not recent such as 22, 27, 40 and 41, reference No. 21 without a date.
5. It is clear what is already known about this topic, the research question is clearly outlined; the research question was justified given what is already known about the topic.
6. The process of subject selection is clear; the variables defined and measured are appropriate, the study methods are valid and reliable; but the sample size seems shortcoming, thus results should be interpreted with caution.
7. The data is presented in an appropriate way, tables are relevant and clearly presented, titles, columns, and rows were labeled correctly and clearly.
8. The text in the results adds to the data, it is a statistically significant result.
9. I am clear about what is a statistically significant result.
10. The results are discussed from multiple angles and placed into context.
11. The conclusions answer the aims of the study.
12. The conclusions were supported by results but no references.
13. The limitations of the study are opportunities to inform future research, future studies should be conducted during the preseason to limit other exercises. Future studies should replicate these findings using larger sample sizes.
14. The study design was appropriate to answer the aim.
15. The article is consistent within itself.

Author Response

We would like to thank you for the detailed and meticulous review along with pertinent comments.

The comments on the revisions are going to be responded item by item. please check for our attachment. 

Reviewer 2 Report

The present study purposed to investigate the effects of plyometric and ankle stability exercises on dynamic balance and lower limb kinetic and kinematic parameters in Taekwondo demonstration athletes with functional ankle instability (FAI). However, despite this manuscript presenting clinical relevance for the injury prevention and rehabilitation fields, as well as adequacy of the scope of the Journal, it still needs improvement, which is considered as a prerequisite to a future publication.

Firstly, I think that the INTRODUCTION of the study can be improved on some points (see specific comments below). The OBJECTIVES are clear. However, no hypothesis was established by the authors. METHODS and RESULTS need some adjustments. A special note is made for some statistical options used by the authors (see specific comments below). The DISCUSSION needs to be expanded at few points and readjusted in others. The CONCLUSIONS need to be revised to return to the objectives of the work. Other considerations are made the following specific evaluation.

TITLE is presented in a clear and concise form, which is consistent to the authors investigation. The authors launch a startling question that draws attention to the reader. I just suggest that authors change to "Effect of plyometric versus ankle…” to make it even more informative

The ABSTRACT includes a brief background and the purpose of the study (to compare the effects of plyometric and ankle stability exercises on dynamic balance and kinetic and kinematic parameters of the lower limb in Taekwondo athletes with FAI). BACKGROUND, METHODS and RESULTS in most part is well written (see comments below). The CONCLUSION is directly related to the objectives.

Line 14 – Please modify to “… and lower limb kinetic and kinematic parameters…”

Lines 16-17 – I suggest authors to invert the order of groups presentation (in order to align with the title).

Line 22 – Please revise the term “anteroposterior side” (anteroposterior axis?? anteroposterior displacement??)

Lines 28-29 – I suggest that authors must revise the following keywords: Taekwondo Demonstration Athlete; Ankle stability Exercise

INTRODUCTION

I think the introduction to the work is well described. The importance of the study is properly highlighted. Authors adequately point out the epidemiological scenario facing injuries in taekwondo athletes. Still, they point to the main biomechanical conditions related to ankle instabilities and the treatment approaches most used in the rehabilitation of these injuries, with an emphasis on plyometric training and ankle sensorimotor exercises. However, there are gaps for reference to some concepts and theories that are not the authors' own (lines 31-33; lines 54-57). These references must be pointed out in the Introduction. 

Line 36 – I believe it is unnecessary to describe the lower limb joints. 

Line 37 - I suggest a paragraph break here, and a junction with the next paragraph. 

Line 38 – I suggest modify to “which creates excessive shock at joints”. 

Lines 40-41 - As mentioned earlier, I suggest adding the paragraphs, but with an adjustment in the wording to avoid repetition of content. 

Line 43 – I suggest modify to “loss a proprioception and neuromuscular changes”. 

Line 44 - I suggest modify to “dynamical changes”. 

Lines 54-55 – Please modify to “could negatively affect sports performance [3–5] and increase the injury risk”. 

Line 61 - I suggest modify to “lower limb kinematic and kinetic parameters”. 

Line 62 - Authors are suggested to point out a set of hypotheses expected for the study. This would strengthen the introduction of the manuscript. 

METHODS

In the most part, the methods and procedures for data collection are described clearly and seem appropriate, in sufficient detail to allow others to replicate and build on published results. The methods comply with ethical principles. The authors pointed out the favorable opinion of the Ethics Committee. The analyses seem in most part appropriate to meet the objectives of the study. Specific considerations are followed.

Line 65 - Authors are suggested to indicate how volunteers are recruited.

Lines 66-67 - Authors are suggested to better define the clinical significance of “ankle instability instrument score >5”.

Line 69 – I suggest that authors must include the assign of and Informed Consent form.

Line 71 - Was there a sample loss in the study? If so, please report it.

Table 1 - I suggest adding the injury time and the ankle instability score, for each group, for better characterization of the sample.

Line 78 – I think the term “on a single leg” is more adequate.

Line 89 - I suggest modifying the subtitle to “Kinematic and kinetic analyses”

Line 99 - I suggest modify to “land on one-limb or land on one-foot”.

Table 2 - Please review the table's indexes for superscript letters.

Line 106 – - I suggest modify to “14 volunteers”.

Line 122 - I think that authors should describe the characteristics of the kinematic acquisition system earlier (not in this subtopic). In addition, they must also describe the acquisition frequency of the force platform.

Line 129 - I suggest modify to “center of pressure (COP) displacement”.

Lines 135 – I don't understand why the Kolmogorov-Smirnov test was used in this case. Sample size is low. Please review it or justify the use of this test.

Line 137 - I suggest that the authors make it clear that the difference between the effects of the interventions will be calculated in the intergroup analysis.

RESULTS

The main results are pointed out in a concise and precise form, according to the norms of the journal. There is no duplication of information in charts, tables and text. The tables used include the requirements necessary to address the main issues.

Line 141 - In fact, I think that what happened was that the increase in CS was higher in the ASE group.

DISCUSSION

The results verified in the present manuscript are at mostly confronted with other previous works. In most part, authors pointed convergences and divergences between the results, and the interpretations of the findings demonstrate ownership in relation to the central theme of the article. Some studies of a similar nature are cited to assist in the foundation of authors’ considerations. However, there are gaps for reference to some concepts and theories that are not the authors' own (lines 172-173; lines 175-176; lines 190-194; lines 231-233; lines 235-237; line 248). These references must be pointed out in the Introduction. The discussion still needs improvements in some key points, as addressed below.

Line 166 - I suggest that the authors start the Discussion by briefly resuming the objectives of the study.

Lines 167-168 - I suggest modify to “after training”.

Lines 180-181 - I believe that authors should consider the differences between baseline values. Do the comparison be made between these baseline values?? This point was not clear to me.

LIMITATIONS

The authors indicate and/or justify possible limitations of the study, both in relation to the aspect of sample selection, and in relation to the technical and methodological aspects.

CONCLUSIONS

Authors’ conclusions are in most part based on the study results and do point to the main purposes of the study.

Line 271 – “…the risks of sport-related injuries reduced”. I think that it is not possible to infer through a cross-sectional study.

Author Response

(The authors gave the same response as above.)
